# Seizure Detection: A Low Computational Effective Approach without Classification Methods

**DOI:** 10.3390/s22218444

**Published:** 2022-11-03

**Authors:** Neethu Sreenivasan, Gaetano D. Gargiulo, Upul Gunawardana, Ganesh Naik, Armin Nikpour

**Affiliations:** 1School of Engineering, Design and Built Environment, Western Sydney University, Penrith, NSW 2751, Australia; 2The MARCS Institute, Westmead, NSW 2145, Australia; 3Translational Research Health Institute, Westmead, NSW 2145, Australia; 4The Ingam Institute for Medical Research, Liverpool, NSW 2170, Australia; 5Adelaide Institute for Sleep Health, College of Medicine and Public Health, Flinders University, Adelaide, SA 5042, Australia; 6Neurology Department, Royal Prince Alfred Hospital, Camperdown, NSW 2050, Australia; 7Central Clinical School, The University of Sydney, Darlington, NSW 2008, Australia

**Keywords:** seizure detection, EEG, feature identification, low computation method

## Abstract

Epilepsy is a severe neurological disorder that is usually diagnosed by using an electroencephalogram (EEG). However, EEG signals are complex, nonlinear, and dynamic, thus generating large amounts of data polluted by many artefacts, lowering the signal-to-noise ratio, and hampering expert interpretation. The traditional seizure-detection method of professional review of long-term EEG signals is an expensive, time-consuming, and challenging task. To reduce the complexity and cost of the task, researchers have developed several seizure-detection approaches, primarily focusing on classification systems and spectral feature extraction. While these methods can achieve high/optimal performances, the system may require retraining and following up with the feature extraction for each new patient, thus making it impractical for real-world applications. Herein, we present a straightforward manual/automated detection system based on the simple seizure feature amplification analysis to minimize these practical difficulties. Our algorithm (a simplified version is available as additional material), borrowing from the telecommunication discipline, treats the seizure as the carrier of information and tunes filters to this specific bandwidth, yielding a viable, computationally inexpensive solution. Manual tests gave 93% sensitivity and 96% specificity at a false detection rate of 0.04/h. Automated analyses showed 88% and 97% sensitivity and specificity, respectively. Moreover, our proposed method can accurately detect seizure locations within the brain. In summary, the proposed method has excellent potential, does not require training on new patient data, and can aid in the localization of seizure focus/origin.

## 1. Introduction

Epilepsy is recurrent sudden events (seizures) caused by abnormal neuronal activity that completely disrupts the brain’s normal functioning [1]. Seizures can lead to partial or complete loss of consciousness, motor–sensory impairment, and, in some cases, death (sudden unexplained death or SUDEP) [2]. Seizures have several underlying causes, ranging from subjective sensations to major motor movements that may lead to injury. According to the literature, epilepsy disorder manifests in childhood and older adults [3]. However, it is also documented that seizures can be caused by brain damage due to stroke, tumours, infections, and neurodegeneration. In other words, the exact etiology of epilepsy remains unknown [4]. A seizure usually lasts from thirty seconds to a few minutes; persistent seizures (status epilepsy) are a medical emergency that can lead to brain damage [5,6].

According to various records, epilepsy affects around 60 to 70 million people worldwide [7] and approximately 250,000 patients in Australia [8]. The major consequence of the disease is the declining quality of life of the patient and the failure of effective throughput due to the unpredictability of seizures. Seizures can be controlled with medication in two-thirds of patients with epilepsy. However, sudden deaths in epilepsy patients are ten times more likely than in the general population [9]. Hence, timely diagnosis and management of recurring seizures are extremely important [4,10].

An expert review can detect most seizures on surface electroencephalography (sEEG) [11]. Inpatient recordings mainly use standard visual analysis of EEG data and consider the patient’s behaviour in the recorded clinical video for a detailed interpretation of seizure events. This clinical analysis is burdensome, expensive, and time-consuming; for instance, the long-term visual examination (usually 24 h of EEG data) may take up to an hour [12]. Given the cost of expert labor (the hourly wage for an EEG technician is USD 28, and a neurologist is USD 134, excluding overhead) and the large testing volume, an hour of expert review of a single recording is very unproductive/expensive. In addition, the promise of new electrode technology for ultra-long-term EEG recordings and current requirements for real-time review has created a desperate need for automated EEG review. Variability in clinical interpretation based on the skill and expertise of the EEG assessor leads to inherent subjectivity [5]. The occurrence of signal disruptions/interruptions, such as physical movements and/or electrical interference, may also render difficult signal interpretation. Several detection methods have been introduced to aid clinicians in this complex/strenuous task, beginning with automatic seizure analysis in the 1970s. Automatic seizure detection uses specific features to identify and categorize seizure events. It involves three main steps (Figure 1): (i) feature extraction, (ii) training and testing of a classifier, and (iii) actual classifier classification [13].

Feature extraction identifies distinguishing signals and removes unnecessary information from EEG data, defining a set of features that can then be processed by classifiers. This approach may flag any deviations from the extracted features as seizures [14]. Feature patterns, such as increased spiking, rhythmicity, or EEG desynchronization are some time-based biomarkers that are employed by the algorithm [15]. Raw EEG signals are temporal (time domain); conversion to the frequency domain through Fourier transforms [16,17], and Wavelet transforms [18,19,20] are commonly used to enhance seizure detection. The advantage of these frequency features is that they are less prone to variations due to electrode location, physical characteristics, relative motion, and three-dimensional contexts [21,22]. Other methods include correlation measurements, the Lyapunov exponents [23], and entropy [24,25,26,27].

In the training stage, the selected features are collected into classes which are subsequently used to classify the EEG signals by using varying machine-learning methods such as nearest neighbour classifiers, support vector machines, artificial neural networks, linear discriminant analysis, regression, and neuro-fuzzy systems to categorize the incidence of seizures [28,29]. Machine learning has grown significantly over the past three decades due to its low-cost, high-speed computing infrastructure, robust algorithm, and large data computing capabilities. The key to the success of any machine learning approach is the availability of large annotated/labelled (by experts) datasets that can be used to extract training features. In EEG seizure classification, this is quite a challenge, thus limiting the performance of training tasks [30]. The number of features is correlated to the complexity of the classifier; when only one feature is used in the classification, it is not as complicated. However, to adequately describe the energy patterns of the seizure, more than one feature is always required [31].

Overall, feature training is repetitive, but the actual classification is a one-time process that is expected to be compatible with new data. Because EEG training and testing involve a large amount of data repetition, the associated time and calculation load are very high. This load may be twice that of standard visual seizure-analysis methods [32]. Classifying patients’ seizures is also highly challenging because of the intermittent, unpredictable dynamics of EEG seizures (spatial and temporal features changing very rapidly).

Moreover, seizure characteristics are patient-specific [33], and long-term training and classifier testing are usually required for effective seizure identification and prediction. This long-term training and classification are a hindrance if the patient is in critical care [34]. Even though advanced machine-learning methods give exciting results in identifying seizures [35], they are not helpful for real-world analyses due to a lack of reproducibility, thus resulting in low sensitivity and specificity in real-world scenarios [12,32]. This is particularly the case if studies aim to detect seizures in a large population without specific individual patient training data. Hence, most clinical settings follow up with the traditional visual analysis of the EEG rather than risking missing epilepsy seizures by any automatic detection method. There is a noticeable gap between traditional (visual) seizure identification and advanced machine-learning methods.

A possible solution for existing seizure-identification methods is providing a concise and accurate EEG feature identification to assist with traditional visual analysis. This paper discusses a new EEG seizure-detection method to overcome the limitations of classifier-based methods while providing a worthwhile upgrade suitable for the existing visual analysis procedure. The proposed method can be employed for multichannel/multimodality analysis and for analyzing individual channels isolated from multichannel data. Section 2 explains the details of the proposed method. The results are presented and discussed in Section 3, and Section 4 concludes the paper.

## 2. Low-Computational Seizure-Detection Method

At the core of the technique is the conversion of the multichannel EEG into a single “feature signal”, the algorithm allows the features of interest to be extracted from the EEG signals, showing only the identified seizure epochs. In the “feature signal”, the seizure presents itself as a baseline discontinuity, creating an “empty space” for the duration of the seizure period. The concept of manipulating the signal to produce a mask that can be used to focus attention only on the features of interest is not new. It is often applied to other physiological signals (e.g., heart-rate detection from electrocardiogram [36]). At the end of the processing, the resulting “feature signal” might not contain visual cues of the original physiological signal; however, this will be fundamental in producing the signal scoring mask. Our seizure detector should not be confused with the so-called amplitude EEG (aEEG) routinely used only for neonate seizure monitoring [37]. However, some of the signal-processing steps involved in our method may seem similar. One crucial difference is that our method processes all the EEG channels installed, regardless of their position, and returns the identified seizure intervals to the original channels for complete spatial analysis of the seizure.

### 2.1. Production of the “Feature Signal”

Epileptic seizures are linked with different electric rhythms. Therefore, EEG signals’ characteristic fluctuations (biomarker) can be used as a pointer to an ictal state [19]. As the EEG signals contain normal physiological rhythms, the difficulty in identifying critical biomarkers prevents the exclusive use of rhythms for seizure detection. Hence, other discriminatory features need to be used to enhance the correctness of seizure identification [38,39,40]. Production of the “feature signal” requires four low-computational steps: signal preprocessing, EEG signals averaging, application of the feature filter, candidate seizure epochs detection, and production of the signals mask. The proposed algorithm’s logical flowchart is shown in Figure 2.

### 2.2. Dataset Details

For this paper, we used a dataset available online through the Physionet platform [41]. This dataset, recorded at the Children’s Hospital in Boston, is from paediatric patients with intractable epilepsy [42]. Clinical experts annotated the dataset, marking the onset and end of each seizure; the annotations have been used as ground truth by our group and other researchers using this dataset. The dataset’s full description and annotations can be found online at https://physionet.org/physiobank/database/chbmit/ (accessed on 10 January 2019). The data comprise multichannel EEG signals collected from 22 paediatric subjects: five male subjects (3–22 years) and seventeen female subjects (1.5–19 years). Subject 21 was evaluated after one and a half years, and due to the large time gap, the case is labelled separately as “chb21”, yielding a total of 23 cases. In December 2010, a new case was included as the 24th case; for this patient age and gender are not known. Each case or subject of the dataset comprises 9 to 42 sequential “edf” files (trials), typically lasting an hour. The files for Case 10 are two hours in duration, while the files for subjects 4, 6, 7, 9, and 23 are four hours each. Records containing at least one seizure are divided into seizure records, and others as non-seizure records. All patients had 3–18 seizures, and 198 of the 684 “edf” records contain seizures. Seven patients had partial-onset seizures, seven had secondarily generalized seizures, and eight had generalized seizures. The beginning and end of each seizure marked by an expert are included in the file summary. EEG recordings of Patient 16 were concise, with an average ictal span of just 8.6 s. Table 1 reviews patient-specific data, such as the number of EEG tests and seizures.

### 2.3. Preprocessing

Each of the 23-channel EEG recordings of the CHB-MIT dataset contains a high level of unwanted noise from physical movements, electromagnetic interferences, and other artefacts. A low-pass filter removes unwanted noise with a frequency cutoff of 35 Hz and an additional power-line noise notch filter. The low-pass filter is the FIR filter applied non-causally (zero-phase shift filter), 50th order, offering an excellent transition band. This pre-filter greatly reduces artefacts and external noises at high frequencies. Moreover, a filter of 50th-order non-causal IIR notch filter at power-line frequency and harmonics; is applied first, and, although it may seem redundant given the low cutoff of the low pass filter, removal of power-line noise is standard in our lab; the code for this filter is supplied to the user, who has the option of turning it off, and in our analysis, it was found to not influence the performances of our algorithm. Users may also want to tweak the central frequency of this filter according to the country where the recordings originate. For this work, we employed a 60 Hz power frequency as the recordings are from the USA.

The last stage of our preprocessing aims to remove any baseline discontinuity (drift) and DC offset of the signals; removing DC and near-DC components is a crucial step for our algorithm, as it requires channel averaging. For this purpose, we employed a recurring notch filter. This methodology restricts the narrow band of frequencies around the DC offset value. A first-order recursive notch filter with a transfer function, $f(z)=\frac{1-z^{-1}}{1-{\lambda z}^{-1}}$, is executed with a λ value of 0.9991. Because of the data sample rate, this value for the parameter equates to a cutoff frequency of ~0.015 Hz. The data used for our evaluation is prerecorded, and this filter was also applied non-causally.

Generalized non-causal filtration is simply achieved by employing the “filtfilt” function included in the Signal Processing Toolbox (Matlab R2020) to avoid phase shifts and simplify the evaluation of results. The cutoff frequencies are quite distant from the typical seizure signal’s central frequency, ensuring that the system can eventually be applied causally and in real-time. The preprocessing process aims to reduce the power of physiological artefacts and remove any signal baseline that could impact the feature filter. As shown in Figure 3 it does not impact the readability of EEG signals.

### 2.4. The Feature Filter

Once the Large DC offset has been removed, and the preprocessing step has reduced significant artefacts, the arithmetic average of all the EEG signals is computed and rectified. The resulting signal is processed by a feature filter, which behaves as a resonator enhancing the seizure components. The filter, which was designed as a 30th-order bandpass FIR filter, in the range of 0.1–4.1 Hz, was split into its high-pass and low-pass components (see Appendix A and codes). The bandwidth of this filter has been devised to assess the Fast Fourier Transform (FFT) content of several seizures compared with the exact lengths of non-seizure EEG. Due to the closeness of the low boundary of our filter to the DC component, which has been, to some extent, attenuated by the preprocessing step, the feature filter produces an analogous output to the simple low-pass filter(this is further discussed in the discussion section). The feature filter aims to increase the passband frequencies’ signal-to-noise ratio. In other words, the EEG is treated similarly to a noisy transmission channel “tuning” a receiver on the desired informative content. The selected frequency range includes most of the oscillation frequency content of the seizure [43].

The resulting signal’s root-mean-square (RMS) envelope is determined by using a 150-sample moving window and then normalized. The moving RMS calculation is assumed to deliver the most comprehensive detail of the magnitude of the feature signal [44]. The length of the RMS calculation is 40% of the sampling period to cover the length of at least one full seizure spike. Normalization ensures that thresholds to determine the seizure epochs can be specified as a fraction of the numeral one in the following step.

### 2.5. Candidate Seizure Epochs Detection

To extract seizure features, we created signal epochs to merge almost contiguous seizure intervals, i.e., separated by a handful of samples. A minimal length of the 1-s epoch (256 samples); this parameter is called seizLinterval in the attached Matlab code (see Appendix A) and needs to be expressed in samples. A second parameter (called seizAggressiveness) is used to exclude the vast physiological artefacts from the RMS power spectrum (i.e., jaws artefacts), which may fall under the seizure spike central frequency. An ideal value for the signal’s power to be considered a seizure was devised in collaboration with an expert (A.N.) neurophysiologist; the ideal value is between 180 and 220, and a smaller value will mark jaw artefacts as a seizure. Along with the frequency-limited power analysis, the lower and upper limits of the resultant feature (decomposed) signal at the point of discontinuity in the DC level are used to discriminate the seizure from other amplitude spikes. The lowest limit is usually between 0.05 and 0.1, and its upper limit is between 0.8 and 1. These limitations avoid the risk of miscalculating amplitude spikes as seizures, thereby creating false alarms. In our algorithm, all of these parameters are passed through a dedicated Matlab function; thus, the user is free to experiment with various bounded values.

### 2.6. Signal Masking

Once the seizure locations are determined, these are applied as a binary mask (1 = seizure) to the signal average. To identify the electrode position/area of the brain from where the seizure is more prominent/originates, we applied the same algorithm to each EEG signal limited to the masked epochs determining which contributes the most to the average. Each channel’s lower and higher amplitudes are compared, and a threshold is selected from the median value of these differences. Each channel’s amplitude level variations are compared against the set threshold. The channel(s) above the threshold is/are then identified as the location originating the seizure. Rectifying the signal before processing is necessary to ensure that the fast oscillations of the seizure dipoles are adequately accounted for in the short RMS extraction.

The seizure algorithm mentioned here only uses a simple logical analysis based on the absolute value of the denoised and filtered data. To further clarify the proposed method, the following section portrays its application to Patient 1, noted as chb01 in the dataset. A total of 42 EEG testing sessions (trials) and seven seizure periods were marked for this patient. Figure 4 shows the application of the proposed algorithm to Trial 1 (file name chb01_01.edf comprising 16 EEG channels); for this trial, there are no seizures and the baseline RMS is contiguous. Instead, application to Trial 3 (chb01_03.edf) shows an apparent discontinuity in the DC level around the seizure event (see Figure 5), forming a visually evident “void” space (discontinuity in the DC level).

### 2.7. Process of Assisted Manual-Aided Seizure Annotation

Figure 4 and Figure 5 show that the proposed algorithm condenses up to one hour of EEG recordings into a single output window. Manual annotation of the visually evident discontinuities (void spaces see Figure 5) can be performed by using a figure point-marking function like the one provided in Matlab, called “ginput” (see Appendix A). Users can also enable zoom and precisely annotate seizure onset and duration. When the duration of discontinuities is too low or so large (less than 0.4 s and greater than 1.8 s), they are not considered a seizure. The manual inspection is easy to detect these variations in no time. We provide an example of this process and its output in Figure 6. Identified seizure sections can then be reported to individual channels (see Figure 7), where brain locations (electrodes) contributing/originating to the seizure can be selected. Each EEG channel has its mean amplitude around the seizures. For the whole 23 channels of the EEG data, the median of that amplitude difference is selected as a threshold to locate the onset of the seizure.

## 3. Results and Discussion

We measured the performance by using the same well-known evaluation parameters to compare our algorithm performance against published seizure-detection methods directly. The evaluation parameters are accuracy, specificity, sensitivity, precision, f-measure, and g-mean. Definitions of these parameters are given below [45]:Accuracy (Acc) = (TP + TN)/Total Samples(1)
Specificity (Spe) = TN/(TN + FP)(2)
Sensitivity (Sen) = TP/(FN + TP)(3)
Precision = TP/(FP + TP)(4)
F-Measure (F_M) = (2 × Precision × Sensitivity)/(Precision + Sensitivity)(5)
G-Mean (G_M) = √(Specificity × Sensitivity)(6)
where TP indicates a true positive, the number of abnormal cases (e.g., epileptic seizure cases) which are predicted as abnormal; FN is a false negative, which is the number of abnormal cases that are predicted as normal; TN is a true negative, which specifies the number of normal cases that is predicted as normal; FP is false positive, representing the number of normal cases that are identified as abnormal by the system [45]. Setting an appropriate statistical threshold on the DC level shift and normalizing the given data show the accurate projection of the location of the seizure. The results for all the EEG test sessions of Patient 1 with the proposed seizure-detection algorithm are presented in Table 2. Out of the 42 total EEG sessions for Patient 1, 49 were TN, and 7 came out as TP.

The algebraic calculations for weighing the performance of the algorithm include sensitivity and specificity, which were calculated for Patient 1 as follows:(7)$$\mathrm{Sensitivity}=\frac{\mathrm{TP}}{\left(\mathrm{TP}+\mathrm{FN}\right)}=\frac{7}{\left(7+0\right)}\;\times \;100=100\%\;(\mathrm{correctly}\;\mathrm{identify}\;\mathrm{the}\;\mathrm{seizures})$$
(8)$$\mathrm{Specificity}=\frac{\mathrm{TN}}{\left(\mathrm{TN}+\mathrm{FP}\right)}=\frac{49}{\left(49+0\right)}\;\times \;100=100\%\;(\mathrm{correctly}\;\mathrm{identify}\;\mathrm{the}\;\mathrm{non-seizures})$$

The results obtained from a detailed CHB MIT EEG dataset investigation are shown in Table 3 and refer to Figure 8a,b. Furthermore, in addition to the visual evaluation, we tested our strategy in automatic mode, without using any classification. The peaks are identified as the output of automation at the point of onset of the seizure (DC level discontinuity), and whenever the duration of such peaks is less than 1.8 s and greater than 0.4 s, it is flagged as seizures; otherwise, it is disregarded. It is easy to ensure that the absence of a classifier does not falsely affect any of the automatic test results. For that, the resultant data of manual and automated analysis are compared so that anyone can understand whether there are any variations between the developed outputs. If false detections have occurred, confirming such variations could take only an additional minute by specifically checking on that section of the resultant waveform. Also, the exact seizure location on the brain where it originated and developed (refer to Table 4) could be identified by the same algorithm process. The use of these compared automatic schemes helps the practising neurologist interpret seizure EEG records more efficiently and accurately. We prepared Table 5 based on such a comparison of automated and manual results and modified the false alarms (less than 0.4 s and greater than 1.8 s) of automated results with the remarks of an expert (A.N). We modified our low computational model with minimum preprocessing stages to generate comparable seizure-detection performance (refer to Figure 8c). Although there are minor variations in the number of erroneous findings, the abovementioned analysis was performed on each patient and test session. This unique seizure-detection concept indicates the possible directions future and existing seizure diagnoses will take. Total sensitivity represents a fraction of acceptable seizure detections in the test data. Specificity calculates a feature vector as a seizure event per hour based on how often the algorithm misclassifies it [46]. More so, the detection rate of false positives and true positives in the analyses of test data is shown in Figure 9.

The proposed algorithm effectively detects 174 of 198 seizure events. The 24 missed seizures reduce the recall or sensitivity parameter of the testing algorithm. There are 46 false-positive findings. The seizure-detection results in Table 3 show the number of seizure events used to test each patient and the successful detections. The results indicate that average diagnostic performances of the visual feature-based seizure-detection technique are reasonable compared to many existing automatic classification methods, with an average sensitivity of 93%, average specificity of 96%, an average accuracy of 94.82%, and an average precision of 81.74% (refer to Figure 8a,b).

On the manual analysis of 24 patients, complete (100%) sensitivity is obtained for 15 patients, while all other patients’ results are above 67%. The poor results (considering sensitivity below 80%) are in patients less than three years of age. The developing brain, especially in infants, differs from the matured brain in fundamental mechanisms of seizure initiation and propagation [47]. This can be explained by the different frequencies seen during a seizure and interictally in infants. In addition, infant data may also contain high levels of artefacts, requiring additional processing outside the scope of this paper. For example, the EEG could be finely processed to remove blinking and jaw artefacts. The latter, in particular, caused false positives in the automated method resolved by the expert (A.N.) consultation. Whenever the duration output peaks (minimum time duration for accurate seizure detection) are less than 1.8 s and greater than 0.4 s, they are flagged as seizures; otherwise, they are disregarded. Excluding the infant data, we see that the average detection sensitivity increases to 96%. The rate of analyzing the true-positive rate in 23 patients is 0.96. This evaluation database comprises 980 h of uninterrupted surface EEG data. The identification rate of false positives per hour of continuous EEG sessions is around 0.04, which is acceptable [48]. Regardless of the sensitivity results, estimating the harmonic mean of the weight of accuracy and sensitivity is imperative. The estimated quantity is called the F-measure, which (shown in Equation (5)) considers all incorrect decisions (both false-positive and false-negative ones). The F-score is generally more powerful than expressing accuracy, particularly for an unbalanced dataset [32]; the average F_M was 85.7%. The geometric mean (Equation (6)) evaluation metric is also adopted in this paper to provide further insight into the accuracy obtained. The average value of G_M is 94.18% (Table 3). The explicit semiology (region of the brain where seizures initiate and progress) is essential for assessing epilepsy [49], so the analysis of such events with the multichannel data amplifies the overall performance (Table 4).

Most erroneous findings may have been due to sudden amplitude changes and a poor signal-to-noise ratio. An additional smooth filter may reduce the erroneous findings, but minimizing the errors due to high amplitude rhythms is difficult.

### Performance Comparison: How It Differs from Existing Methods

Seizure detection is challenging, and real-time detection is even more difficult. The new algorithm allows seizure detection by using long-term continuous EEG recordings. Therefore, this section compares our proposed method’s performance with other important methods in the literature that correspondingly evaluate long-term EEG. Since our method does not have a classifier, this should be considered a general performance comparison. The majority of the past literary works used unique EEG data to train the classifier [50,51,52]. The proposed algorithm herein is a simple method that does not involve classifier training, thus making the analysis a less complex computation. The discussed method attains a positive-detection sensitivity of 93%, equivalent to Shoeb and Guttag’d [53] algorithm results. Many previous studies have found that automatic seizure detection performs better, as shown in Table 6. However, these studies involved a large volume of training on data from previous seizures (using patient-specific methods) or selective practice on the duration of different seizures and non-seizure (using seizure-specific methods).

Most of the published methods in the literature use a support vector machine (SVM) classifier to detect seizure events [54]. A patient-specific seizure-prediction method is used [55] to optimize the SVM classifier in the EEG recordings from the CHB dataset. Shoeb et al. extracted the frequency and three-dimensional features of the data and then combined non-EEG features to produce a feature vector; an SVM classification followed this. The latter method was identified in an event-based assessment. The results show a sensitivity of 96% and an accuracy of 96%, with a false-assessment rate of 0.08 per hour. An analogous study used five patients’ records from the CHB-MIT dataset containing 65 seizures. The seizure durations were evaluated by using a linear discriminator classifier [56]. They found 83.6% sensitivity, 100% specificity, and 91.8% accuracy. While the results are promising, the approach is completely patient-specific and cannot be generalized to other patients.

Moreover, the classifiers mentioned above failed to detect non-seizure periods (with many false positives) accurately. A general seizure-detection system was developed to estimate the interrelationships within a network of interested power-band groups [46]. The algorithm showed an average sensitivity of 83% and an average incorrect-detection rate of 2.9%. Other researchers have suggested several ways to detect epilepsy seizures with varying results (Table 6). A segment-based seizure analysis obtained a sensitivity of 91.72% and a specificity of 94.89% in Reference [57]. This method works with the ELM algorithm on a neural network. The same study’s event-based analysis showed a sensitivity of 93.85% and a false detection rate of 0.35/h.

Samiee et al. [58] used a seizure analysis 2D plot of EEG and a textured grey image to extract multivariate features to differentiate seizure events from other events. This method achieved a low sensitivity of 70.19% and a specificity of 97.74%, using a stochastic gradient descent (SGD–SVM) classifier. Employing the dynamics of EEG signals and the time-delay embedding method of high dimensional space, a sensitivity of 88.27% and specificity of 93.21%, are being achieved [59]. An improved particle swarm optimization (IPSO) and neural network classifier study conducted by Nasehi et al. [60] achieved a sensitivity of 98% with a low latency period and a false-positive rate of 0.125/h. The approach has the capacity for practical analysis without the convergence of seizure parameters. Supratak et al. [61] showed that using stacked autoencoders and logistics class, a sensitivity of 100% is possible with a low error rate. However, this study was limited to a small set of cases of the CHB dataset.

The seizure-detection methods reviewed (Table 6) above use advanced machine-learning techniques to extract seizure indicators from EEG signals. Their performance depends on the choice of hyperparameters and data and requires a degree of proficiency and considerable effort. As our proposed method is not fully automatic, and we performed the traditional visual analysis for seizure detection, the performance comparison is partially appropriate with existing automated methods. However, for comparison, we selected the literature based on the same dataset. Moreover, seizure detection is a patient-specific method; all advanced machine-learning algorithms are trained offline, using a dataset, and then implemented in real-time for accurate seizure identification. This method is always time-consuming and demands bulk data for initial training and testing—the higher the volume of data used in training, the more accurate the results that are obtained in classification. No existing algorithms can perform without training, and applying them to a naive patient’s EEG is impossible. The proposed study does not utilize a training section, so it can be implemented in real-time, without any offline training. Many patients do not have training data available, e.g., patients in intensive-care units.

**Table 6 sensors-22-08444-t006:** Performance comparison of our proposed method with previous studies in the literature.

Author	Classifier	Sensitivity Percentage	False-Detection Rate
Shoeb [55]	SVM	96%	0.08/h
Samiee et al. [58]	SGD–SVM	70.19%	-
Khan et al. [56]	LDA	83.6%	-
Mansouri et al. [46]	PBI, DN, CN	83%	2.82/h
Zabihi et al. [59]	LDA	88.27%	3.04/h
Hunyadi et al. [62]	NNL	81%	0.11/h
Yuan et al. [57]	NNL	93.85%, 91.72%	0.35/h
Nasehi and Pourghassem [60]	ANN	98%	0.125/h
Supratak et al. [61]	SAE	100%	-
Proposed Method (Manual/Automatic)	-	93/88%	0.04/h

Moreover, the volume of data required in automatic training and a physician’s visual analysis workload is relatively large. The use of multichannel data makes seizure identification difficult. Shrinking the size of the analysis of multichannel data to a single datum reduces the time needed and the possible complications for seizure analyses. The raw EEG database’s size is around 32.3 GB; after averaging the signals, the total size is only 1.28 GB (~1/25th of the original size). This significant data-volume reduction allows for faster seizure-algorithm implementation. As an additional benefit, the visual evaluation time is shortened, reducing the number of screens/figures necessary for the evaluation; our method proved suitable for 1-h datasets to be assessed at once rather than the typical few seconds per page.

## 4. Conclusions

As seizure detection is a challenging procedure, real-time detection with classification is difficult. A large and diverse amount of information is required for every classification method. Moreover, because of the general nature of seizures, they can vary from person to person. So, automation using classifiers is always a patient-specific method that involves training a large dataset over an extensive period. With a dataset such as the CHBMIT used for this study, it is difficult to divide the dataset into balanced training and testing sets; hence, performances are usually assessed by employing a leave-one-out approach and/or cross-validation. Exclusively, our algorithm avoids all of these complexities and allows for long-term continuous EEG recordings for seizure detection. The highlight factors of our method are the innovative approach of using whole EEG data to detect seizures without any complex classifiers and the less time required for the data analysis.

As classification approaches often require subject-specific training and folding of results to maintain the claimed performances, our proposed method is pure detection, which does not require training to be used on new patient data. It also detects the level of brain areas/lobe or single electrodes if desired. It is possible to infer from Table 6 that our methods perform well against state-of-the-art classification methods for which, on average, the sensitivity percentage performs at 88.56%; our method scores 88% for fully automated detection and 93% for assisted manual detection. Our results (Table 3 and Table 5) show that, while our method works exceptionally well on some patients, further tweaking may be required to minimize false detection, mainly if the general EEG SNR (i.e., movement artefacts) is low. There are three patients with precision scores below 60% with the proposed approach (see Table 3). Those are Patients 4, 7, and 13; for two (Patients 4 and 7), our specificity and sensitivity are still relatively high, while they fall for a patient. Furthermore, our evaluation was made based on a single free available dataset. A further clinical investigation may be required to demonstrate its full capability.

The lack of various datasets annotated with seizures for adult patients makes it hard to show the extensive validation of our algorithm. Overall, the method is not perfect regarding the results for the available CHBMIT dataset, and it is robust in action to the variation in electrode position. The records provided with the dataset state that the EEG recordings were taken by using the International 10–20 System of electrode locations and terminology. However, 17 files have seizures with different channel locations from the remaining files. The proposed method’s overall automated performance affected three patients: Patients 6, 12, and 13. Avoiding these outliers increases the overall TPR to 97%. Excessive artefacts are the main reason for the reduced performance in these patients. The recordings of the EEG signals for Patients 6 and 13 contain significant artefacts, and an improved preprocessing method is required to distinguish ictal events. Some artefacts, such as muscle artefacts, ocular artefacts, blinks, etc., have well-known patterns and automated erasing methods, which we have not explored for this paper. We suggest that adaptive or Weiner filtering artefact removal methods will be more compatible with the proposed algorithm.

Furthermore, Patients 2, 4, 11, and 23 were shown to have low sensitivity in the automated analysis because of a few missed detections. These detections were missed due to their brief and segmented duration requiring tailored adjustment of the algorithm parameters (pks, locs, and width in Matlab code, see Appendix A). The manual visual analysis gives more authentic results in a short time compared to automated results. Further extensive analysis is undoubtedly required to identify a feasible approach to avoid missed detections in the generalized automation approach.

In conclusion, our algorithm is a significant change in concept for the research community and can serve as a synergistic tool to enhance their current processes. Future work includes the integration of localized seizure detection, improved signal denoising, and interactive parameter sweep to reduce the missed detection of brief seizures. Furthermore, this work could be a stepping stone for more robust feature extraction tailored to specific patients.

## Figures and Tables

**Figure 1 sensors-22-08444-f001:**
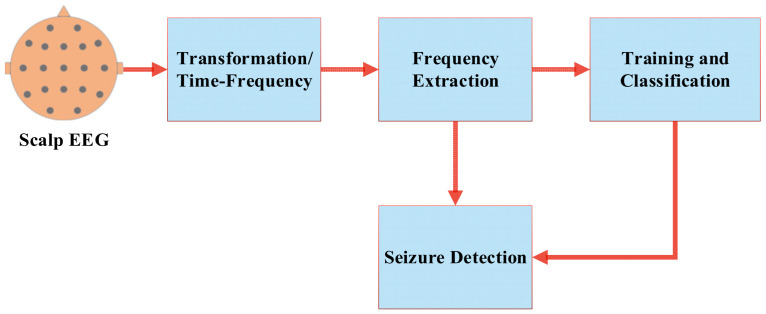
Common stages of automatic seizure-detection process based on feature extraction and classification.

**Figure 2 sensors-22-08444-f002:**
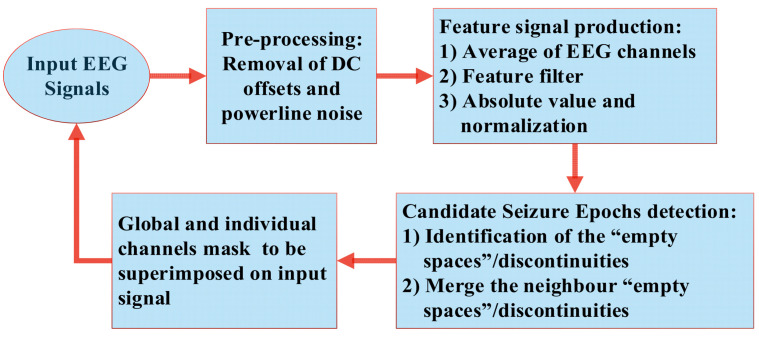
Logical flowchart of the proposed seizure-detection algorithm. The loop is repeated once for every dataset of one hour in length. An expert can review the results of both manual and automated data to identify any variations in false alarms and make corrections within a few minutes.

**Figure 3 sensors-22-08444-f003:**
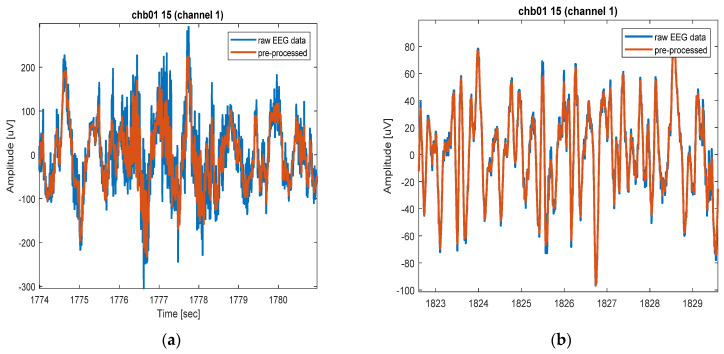
Direct comparison of raw EEG (blue trace) and preprocessed EEG signal. (**a**) Direct comparison of raw EEG (blue trace) and pre-processed EEG signal during artefact. (**b**) Direct comparison of raw EEG (blue trace) and pre-processed EEG signal artefact clean section.

**Figure 4 sensors-22-08444-f004:**
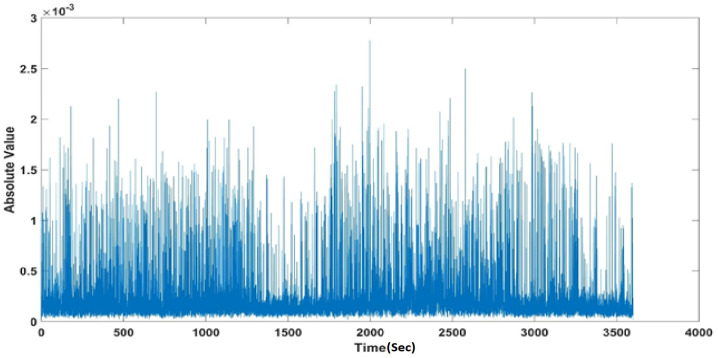
Output for chb01_01, using the proposed seizure-detection algorithm, shows no seizure activity.

**Figure 5 sensors-22-08444-f005:**
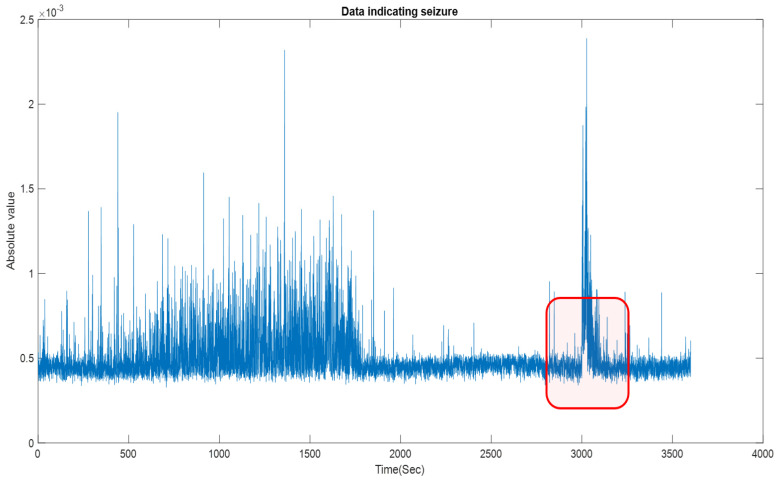
Feature extraction showing the onset of seizure for chb01_03, using the proposed seizure-detection algorithm. The highlighted red box shows the region of seizure.

**Figure 6 sensors-22-08444-f006:**
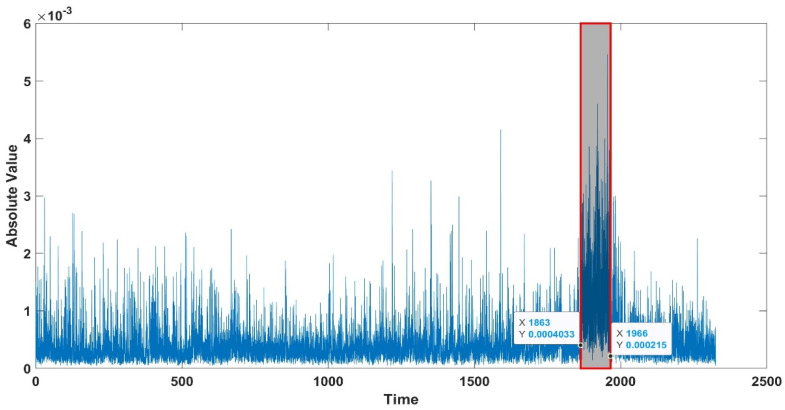
Feature showing the onset of seizure marked in the output window for chb01_26, using the proposed seizure-detection algorithm.

**Figure 7 sensors-22-08444-f007:**
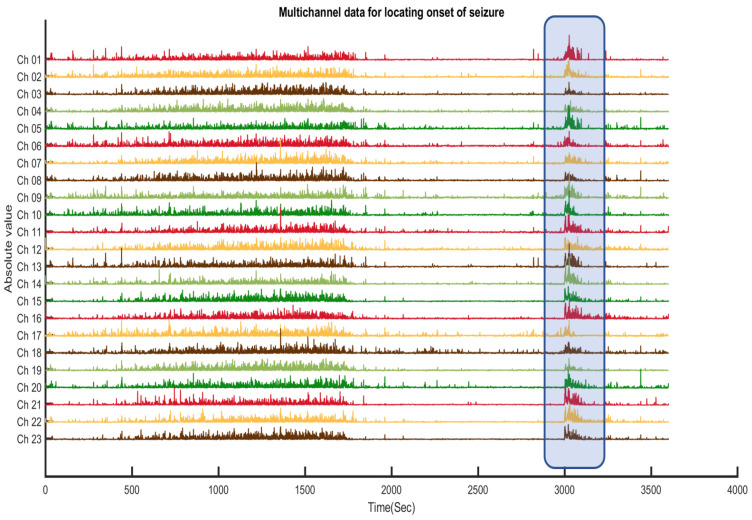
Location identification for chb01_03 using the proposed algorithm. Channels 1, 5, 9, and 14 show the amplitude level above the set threshold corresponding to the frontal-lobe seizure.

**Figure 8 sensors-22-08444-f008:**
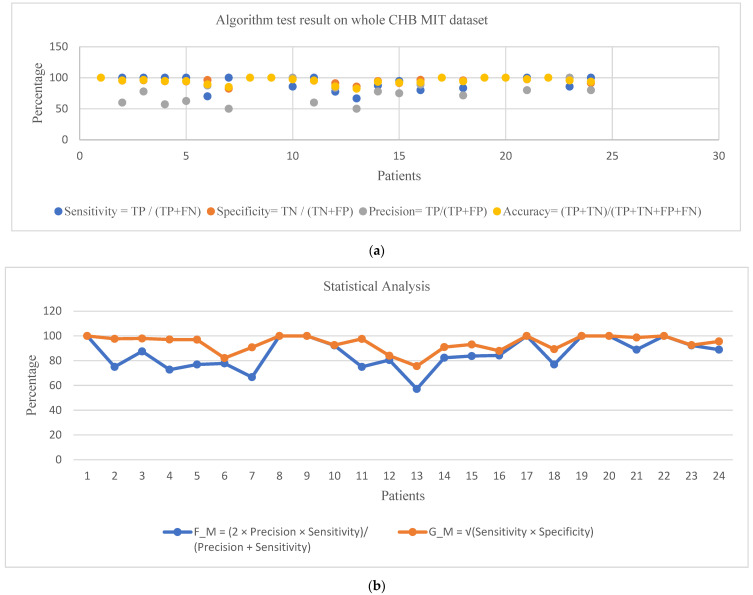
(**a**) Test results for the whole CHB MIT EEG dataset. (**b)** Test results for the whole CHB MIT EEG dataset (F_M is F-measure; G_M is geometric mean). (**c**) Automated test results for the whole CHB MIT EEG dataset.

**Figure 9 sensors-22-08444-f009:**
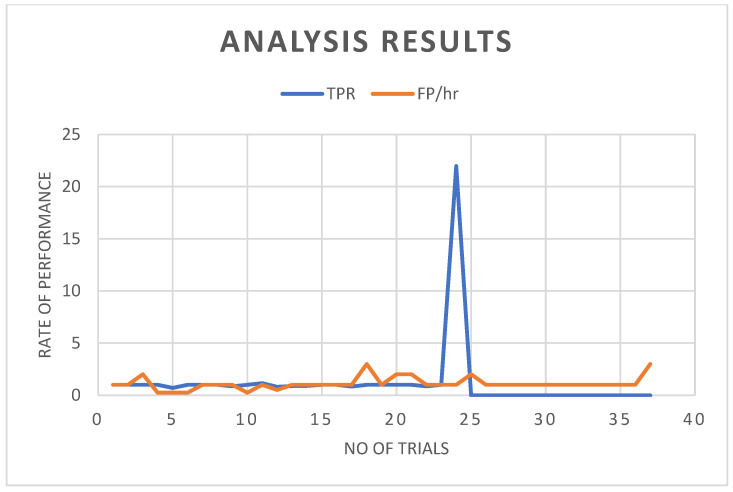
Time-based performance of the proposed algorithm: (TPR is total positive rate) and (FP/hour is false positives per hour. Note: The figure shows the total positive and false-positive rates per hour for all the individual trials of patient data. In the dataset, a single patient may have a varied number of 15 to 42 data files. Each data file is named according to the number of trials; that is, Patient 1 has 42 data files or trials.

**Table 1 sensors-22-08444-t001:** CHB MIT dataset.

Patient	Age/Gender	No. of EEG Tests	No. of Seizures Annotated
chb01	11/F	42	7
chb02	11/M	36	3
chb03	14/F	38	7
chb04	22/M	42	4
chb05	7/F	39	5
chb06	1.5/F	18	10
chb07	15/F	19	3
chb08	3.5/M	20	5
chb09	10/F	19	4
chb10	3/M	25	7
chb11	12/F	35	3
chb12	2/F	24	40
chb13	3/F	33	12
chb14	9/F	26	8
chb15	16/M	40	20
chb16	7/F	19	10
chb17	12/F	21	3
chb18	18/F	36	6
chb19	19/F	30	3
chb20	6/F	28	8
chb21	13/F	33	4
chb22	9/F	30	3
chb23	6/F	9	7
chb24	NA	22	16

**Table 2 sensors-22-08444-t002:** Algorithm test result for Patient 1, dataset chb01.

Patient	Age	Gender	No. of Seizures Annotated	Seizures Annotated	Seizures Detected	Result	Duration of Seizure
1	11	F	7	chb01_03.edf	chb01_03.edf	TP	3000–3040
				chb01_04.edf	chb01_04.edf	TP	1467–1510
				chb01_15.edf	chb01_15.edf	TP	1730–1772
				chb01_16.edf	chb01_16.edf	TP	1015–1071
				chb01_18.edf	chb01_18.edf	TP	1720–1812
				chb01_21.edf	chb01_21.edf	TP	326–421
				chb01_26.edf	chb01_26.edf	TP	1862–1970

**Table 3 sensors-22-08444-t003:** Algorithm test results for the whole CHB MIT EEG dataset.

Patient Age/Gender		No. of Trials	No. of Correct Seizure Detections (TPs)	No. of False Positives (FPs)	No. of False Negatives (FNs)	No. of Correct Non-Seizure Detections (TNs)	Sensitivity = TP/(TP + FN)	Specificity = TN/(TN + FP)	Precision = TP/(TP + FP)	Accuracy = (TP+TN)/(TP + TN + FP + FN)	F_M = (2 × Precision × Sensitivity)/(Precision + Sensitivity)	G_M = √(Sensitivity × Specificity)
1	11/F	42	7	0	0	49	100.00	100.00	100.00	100.00	100.00	100.00
2	11/M	36	3	2	0	41	100.00	95.35	60.00	95.65	75.00	97.65
3	14/F	38	7	2	0	47	100.00	95.92	77.78	96.43	87.50	97.94
4	22/M	42	4	3	0	49	100.00	94.23	57.14	94.64	72.73	97.07
5	7/F	39	5	3	0	47	100.00	94.00	62.50	94.55	76.92	96.95
6	1.5/F	18	7	1	3	26	70.00	96.30	87.50	89.19	77.78	82.10
7	15/F	19	3	3	0	14	100.00	82.35	50.00	85.00	66.67	90.75
8	3.5/M	20	5	0	0	15	100.00	100.00	100.00	100.00	100.00	100.00
9	10/F	19	4	0	0	23	100.00	100.00	100.00	100.00	100.00	100.00
10	3/M	25	6	0	1	31	85.71	100.00	100.00	97.37	92.31	92.58
11	12/F	35	3	2	0	40	100.00	95.24	60.00	95.56	75.00	97.59
12	2/F	24	31	6	9	61	77.50	91.04	83.78	85.98	80.52	84.00
13	3/F	33	8	8	4	48	66.67	85.71	50.00	82.35	57.14	75.59
14	9/F	26	7	2	1	35	87.50	94.59	77.78	93.33	82.35	90.98
15	16/M	36	18	6	1	64	94.74	91.43	75.00	92.13	83.72	93.07
16	7/F	20	8	1	2	28	80.00	96.55	88.89	92.31	84.21	87.89
17	12/F	23	3	0	0	24	100.00	100.00	100.00	100.00	100.00	100.00
18	18/F	36	5	2	1	44	83.33	95.65	71.43	94.23	76.92	89.28
19	19/F	30	3	0	0	33	100.00	100.00	100.00	100.00	100.00	100.00
20	6/F	28	8	0	0	38	100.00	100.00	100.00	100.00	100.00	100.00
21	13/F	33	4	1	0	38	100.00	97.44	80.00	97.67	88.89	98.71
22	9/F	30	3	0	0	33	100.00	100.00	100.00	100.00	100.00	100.00
23	6/F	9	6	0	1	16	85.71	100.00	100.00	95.65	92.31	92.58
24	-	22	16	4	0	42	100.00	91.30	80.00	93.55	88.89	95.55

Average Sensitivity = 93% (correctly identify the seizures), Average Specificity = 96% (correctly identify the non-seizures), Average Precision = 81.74%, Average Accuracy = 94.82%, Average F_M = 85.79%, Average G_M = 94.18%.

**Table 4 sensors-22-08444-t004:** Location of seizure.

Patient	Seizure Location
chb01	Frontal
chb02	Temporal
chb03	Frontal
chb04	Temporal
chb05	Frontal
chb06	T/Occi
chb07	Temporal
chb08	Frontal
chb09	T/Occi
chb10	Temporal
chb11	Temporal
chb12	Temporal
chb13	T/Occi
chb14	Frontal
chb15	Temporal
chb16	Temporal
chb17	Temporal
chb18	Frontal
chb19	Frontal
chb20	Temporal
chb21	Temporal
chb22	Temporal
chb23	Temporal
chb24	Temporal

There are no detailed expert annotations available on seizure locations in the CHBMIT dataset.

**Table 5 sensors-22-08444-t005:** Automatic test results for selected patients.

Patient Age/Gender		No. of Trials	No. of Correct Seizure Detections (TPs)	No. of Correct Non-Seizure Detections (TNs)	No. of False Positives (FPs)	No. of False Negative (FNs)	Sensitivity = TP/(TP+FN)	Specificity = TN/(TN+FP)
1	11/F	42	6	48	3	1	85.71	94.12
2	11/M	36	2	38	3	1	66.67	92.68
3	14/F	38	7	45	0	0	100.00	100.00
4	22/M	42	2	44	2	1	66.67	95.65
5	7/F	39	5	44	1	0	100.00	97.78
7	15/F	19	3	21	1	0	100.00	95.45
8	3.5/M	20	5	25	0	0	100.00	100.00
9	10/F	19	3	22	0	0	100.00	100.00
10	3/M	25	6	31	1	1	85.71	96.88
11	12/F	35	2	37	3	1	66.67	92.50
17	12/F	23	3	26	0	0	100.00	100.00
18	18/F	36	5	41	1	1	83.33	97.62
19	19/F	30	3	33	0	0	100.00	100.00
20	6/F	28	5	33	2	1	83.33	94.29
22	9/F	30	3	33	1	0	100.00	97.06
23	6/F	9	2	11	0	1	66.67	100.00
24	NA	22	15	36	2	1	93.75	94.74

Average Sensitivity = 88% (correctly identify the seizures), Average Specificity = 97% (correctly identify the non-seizures).

## Data Availability

Please visit <https://physionet.org/content/chbmit/1.0.0/, accessed on 10 January 2019>.

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
