# Peer review of "Seizure Detection: A Low Computational Effective Approach without Classification Methods"

_sensors, 2022, doi:10.3390/s22218444_

Round 1
Reviewer 1 Report
1-Make sure your figures do not need any reference
2- Add the following papers to your literature review and compare your results with it
- Epileptic seizures detection using deep learning techniques: a review
3-what is your innovation in this paper?
4-some sections of the paper need references
Reviewer 2 Report
Thank you for giving me the opportunity to review this interesting manuscript. In this manuscript authors present a new algorithm to detect seizures. It is quite relevant due to the time that this can require.
In line with the aim of the paper is my major concern. I did not find the exact algorithm into the manuscript. I would like that authors report the MATLAB script that allow them to extract these data. It would be necessary to replicate the analysis as well as for clinicians to conduct it. In this line, most of the clinicians do not have knowledge about computer programming or developing MATLAB´s script. At this point, it is crucial to provide them directly and easy information. Thus, it would be appreciate if the manuscript include the requirements and instrumental to run the script (for instance, MATLAB with XXX extensions).
I truly believe that reporting the exact script into the manuscript or as a supplementary material will significantly improve the importance and relevance of your publication. I made this recommendation since I found quite interesting the method you presented.
Minor points
Line 1. Include only your type of article (research article).
Line 20. “To identifying” should be “to identify”.
Line 164. “seventeen” instead “Seventeen”.
Line 325 and line 337. A space before the reference is needed. Please check the document in order to correct the same typo.
Line 338. It seems that the words are written in bold. Please correct.
Regarding figures, I recommend to place them close to the first mention. However, due to the methodological character of the manuscript, it is up to you.
Reviewer 3 Report
The authors aimed to develop a novel EEG based epilepsy detection system that minimizes the practical difficulties of other previously proposed automated systems. Broadly, I think the work targets a meaningful space and as presented has premise, however, my major concerns with the present work are:
1. The specific process of the proposed system is unclear and not described in a manner that could be accurately replicated, and;
2. The authors do not really directly discuss the precision of their system, which on average is okay but not great and varies quite widely (SD ~ 17.5%, range = 50 – 100%).
There are a few minor spelling mistakes, and grammatical and syntactical errors that need to be corrected, but they currently do not overly hamper the reading of the manuscript. My more specific comments are as follows:
1. In reference to expert review of EEG, the authors write: “This clinical analysis is burdensome, expensive, and time-consuming. For instance, the long-term visual examination (for instance, 24 hours of EEG data) may take up to an hour.”
1a. In the grand-scheme diagnostics, is an hour of expert effort really that time-consuming and/or burdensome?
2. The authors write: “The occurrence of signal artefacts such as physical movements and electrical interference may also make interpretation difficult.”
2a. I think the authors need to elaborate upon this more. Online filters are not uncommon and will pretty readily deal with electrical interference. Similarly, depending on the specific routine for the EEG investigation of epilepsy physical movement is largely expected and/or minimized
3. The authors write: “making very in-depth reports as a study model to develop a training model is quite a challenge, as there requires a detailed mapping of primary EEG seizure events.”
3a. I would like to see the authors elaborate on this. It is fundamental to their criticism of machine-learning based approaches, yet is sparse in the detail and is the method used for developing the proposed system (although I do note that the system does not appear to rely on the visual annotations of the data).
4. The authors write: “Even though advanced machine learning methods give exciting results in identifying seizures [34], they are not useful for real-world analysis due to a lack of reproducibility, low sensitivity, and specificity.”
4a. Additional citations and/or evidence need to be provided for this claim.
5. Why was a pediatric sample used, and how would one expect the changes in neural structure and connectivity that occur across childhood/adolescence to be accounted for in the automated detection procedure?
6. The participant data has a pretty wide spread of ages, and more importantly developmental phases. For example, 5 individuals were infants (3 years or less) while 3 were adults (18+).
6a. There are notable differences in brain activity and the symptom profile of epilepsy between these phases, how does this system handle these?
7. Was the clinical annotation of the EEG datasets for the on- and off-set of each seizure associated with behavioral recordings, or just purely based on the visual EEG inspection?
8. What was the trajectory of the 7 patients who had partial-onset seizures? Did they remain partial, or did they generalize?
8a. What were the patient specific true/false positive/negative rates? Did the type of seizure make a difference?
9. The text reports that 198 recordings contained seizes from 22 patients, but Table 1 indicates that seizures were recorded for all 24 patients?
9a. Which is correct?
10. The authors should report on the electrode locations of the recordings. They report that were 23 channels, but not their distribution or location. Assumedly, it was a symmetrical placement across the entire head, but were they equivalent across participants?
11. What exactly are the transition band features of the first FIR bandpass filter? Given the chosen frequency cut-offs the chosen order of the filter seems low, and I’d be concerned about the features relevant to the high-pass threshold.
12. What exactly is the purpose of having a notch filter with a chosen band equivalent to the high-pass band from the FIR bandpass filter, and the high-pass band of the second FIR bandpass filter?
12a. I’d be very interested to see some kind of signal quality analysis before and after that step, and how much it removes baseline drift in comparison to the previous bandpass.
12b. How different is the data classification if you just run the 2-Zeros 2-Poles bandpass FIR filter?
13. The authors write: “The selected frequency range includes most of the oscillation frequency content of the seizure [40].”
13a. The cited paper refers to high-frequency oscillations in a broad 80-500 Hz range, while the authors refers to a 0.1 – 4.1 Hz band. Can the authors clarify their chosen frequency band and the reason why they selected it?
14. To clarify, EEG data are “condensed” across all channels simply by averaging and relying on wave summation properties to generate a “single” channel of information. This single channel data is then analyzed for temporal information about the onset of a seizure. When found, the RMS procedure is then applied to all channels.
14a. Is this correct?
14b. How effective is that method for non-generalized seizures (7 out of 23 patients) or focal-seizure patients (0/23). Are there concerns that the averaging step may obscure those waveforms?
15. Why was an absolute value chosen, rather than allowing the directionality of the data and representing the dipole?
16. The authors write: “The moving RMS calculation is regarded to deliver the most comprehensive detail of the magnitude of the EEG signal.”
16a. This needs to be elaborated upon, as well as cited.
17. The process of what the RMS moving window is doing needs to be better described and further explained.
17a. What is the movement step for the window? 1 second, 150 seconds, 75 seconds, etc.
17b. How is the “seizure location” determined? Does the process compare the calculated RMS value to a pre-determined threshold, or compute the difference between successive windows, and compare this difference to some form of threshold?
17c. Further, is the 2.5-minute window sufficient? As noted by the authors brain changes occur on a msec scale, and some ictal events can be short (on the sec scale) as seen in patient 16.
18. After expanding the algorithm to the whole channel data, the authors write that: “each channel’s lower and higher amplitudes are compared, and a threshold is selected from the median value of these differences. These variations in the amplitude level of each channel are compared against the set threshold.”
18a. This is very unclear, it seems to suggest that there are multiple threshold values. But I think there is only one.
18b. Is this suggesting that for each channel a maximal difference value is calculated, then the median of these difference values is set as a threshold. That threshold is then compared against the channel specific difference value?
18bi. Is that not circular in its logic? 50% of the channels will inherently be above the median?
19. At the end of page 6 and beginning of page 7, the authors appear to describe a number of MATLAB specific functions that provide utility to the proposed system.
19a. Can the authors clarify the meaning of these descriptions? Are they specific parts of the detection procedure?
20. The authors refer to the use of MATLAB’s ‘ginput’ function, which from my understanding and the authors description requires a manual selection of data.
20a. The authors should specify that at this point, the proposed system is no longer automated, and is relying on visual annotation.
20b. Also, at what point in the process does this occur, is this visual inspection still on the ‘single channel’ collated data or the re-expanded whole channel data. Please clarify.
21. The authors write: “Setting an appropriate statistical threshold on the DC level shift and normalizing the given data shows the accurate projection of the location of the seizure.”
21a. This line seems to refer to the success of the proposed system, prior to the presentation of any specific evaluation metric. Can the author clarify?
21b. Further, I think the authors would do well to explicitly describe the moving RMS window as the normalization procedure.
22. In paragraph 2 of page 8 the authors describe the efficacy of the proposed algorithm, noting that 174 of the 198 annotated seizures, 23 false negatives, and 46 false positives were detected.
22a. Should this be 24 false negatives?
23. The ratio between true and false positives is 1:3 (~25% false positives) which seems high, however looking at Table 6 it appears that most of this is clustered in younger individuals. Does this perhaps suggest that the system is not good for young children and/or the type of epilepsy they have?
23a. How do these performance metrics compare between types of epilepsy?
24. The authors write: “The explicit semiology (region of the brain where seizures initiate and progress) is important for assessing epilepsy [44], so the analysis of such events with the multichannel data amplifies the overall performance (Table 5).”
24a. Table 5 just presents a “region” of epilepsy for each patient. How exactly is it relevant to this statement, and/or the proposed algorithm? Were these locations determined from the analysis, how do they compare to the expert annotations?
25. The authors write: “Most of the erroneous findings may have been due to sudden amplitude changes and poor signal-to-noise ratio.”
25a. Did the authors compute signal to noise ratios for each recording? If so, they may wish to report these or at least an average per participant.
25b. Also, they may want to elaborate no why sudden amplitude changes are potentially erroneous. It could very easily be suggested that such changes may be directly related to the increased neural synchronization characteristic of epilepsy.
26. In their performance comparison, the authors write: “However, these studies involved a large volume of training on data from previous seizures (using patient-specific methods) or selective practice on the duration of different seizures and non-seizure (using seizure-specific methods).”
26a. This really needs to be further explained. Why exactly is this problematic, especially as the previous line states that automatic seizure detection demonstrates better performance.
27. Again the authors write: “While the results are promising, the approach is completely patient-specific and cannot be generalized to other patients.”
27a. Why is this a problem? Precision/Individualised medicine has seen a huge push and can often lead to better care. Elaborate on the issue here.
28. What is the purpose of Table 6? Additionally, it is noted referenced in text.
29. Figure 7 needs to be accompanied by an explanatory caption.
30. The authors write: “None of the existing algorithms can perform without training, and it is not possible to apply them to a naïve patient’s EEG.”
30a. The authors need to further explain why this is an issue. The used dataset has many hours of data per person, that could readily be used for training (indeed a number of the papers they compare their performance to, did exactly that).
31. The authors write: “As an additional benefit, the visual evaluation time is shortened, reducing the number of screens/figures necessary for the evaluation.”
31a. Given that in the authors proposed system, whole channel data is still visually assessed, the authors need to clarify exactly how the visual inspection of EEG data is improved..
32. Figure 6 needs to better identify the channels that cross the amplitude threshold and the threshold itself. The current colour coding is obscuring the value of the image.
33. Figure 7 needs a more detail figure caption. What exactly is ‘no of trials’ for the X axis? How was this determined?
34. Section 5 has two paragraphs that start with the phrase ‘In conclusion…’ This sections needs to be rewritten for clarity.
35. In their conclusion, the authors write “…further tweaking may be required to minimize false detection…”
35a. If the authors are aware of this being an issue with the proposed system, why was this not addressed? It does not seem a great choice for the authors to actively criticise their system in their own conclusion.
Reviewer 4 Report
Reviewer #: 1
Manuscript Reference Number: sensors-1740238
Title: Seizure detection: a low computational effective approach without classification methods
Journal: Sensors
Authors: Neethu Sreenivasan, Gaetano Dario Gargiulo, Upul Gunawardana, Ganesh Naik, Armin Nikpour
Summary: The authors claim that they found a simple way to detect seizures by applying a combinations of frequency filters, RMS, and median to scalp-recorded EEG data. Although the calculation process is simple, the detection performance is comparable to other reported methods.
Evaluation: I like the idea of the study. Sometimes our eyes can simply see the presence of seizure onsets in the multivariate time-series data visualizations without running computationally expensive non-linear approaches. I actually personally wonder if those complicated machine learning algorithms can beat human-generated ‘ground truth’ because of the logical circularity of the definition of the ground truth (ha ha). The overall impression of the paper is just ok. It could be improved, I believe. Let me list the points I found that can be problematic/improved.
· I’m not a big fan of the showing around tables with numbers repeatedly in a paper. It’s better reported by using a reasonable plotting such as a violin plot or at least a box plot. But I know on the other hand sometimes medical doctors do this, so maybe this is a difference of practices. I still don’t see the point though.
· There are numerous small errors. What is TPR? The initial appearance of this acronym is not properly defined. What is aEEG? Is this a typo?
· ‘Low computational seizure detection method’ should be something like ‘Seizure detection with computationally demand’
· The Figure 2 is wrong. If No, the loop never ends.
· What’s the sampling rate of the data? Without reporting it, reporting the FIR filter order (as 50th) does not make sense. Anyway, for 0.1-50Hz band-pass, FIR with model order of 50 sounds very short. Are the authors sure that 0.1-Hz high-pass was applied with model order of 50?
· The reason for the repetitive use of frequency filter is not clear to me. If the low-frequency data below 0.1 Hz needs to be suppressed well, the authors can apply the band-pass separately with high-pass and low-pass with aggressive high-pass i.e. using a narrow transition bandwidth only for the high-pass. I also wants to see the frequency profile of the 2-Zeros 2-Poles bandpass FIR filter. I’m not a telecommunication engineer so I don’t know what the authors are talking about. Please make effort to communicate.
· Why is using 150-s RMS filter better than not using it? This kind of arbitrariness in the choices of the approach should be explained better. I guess this is more like the authors’s heuristics, but then please describe so clearly.
· The abstract does not explain the algorithm at all although the authors say it is a simple approach. If it is simple, please report it in the abstract.
· Figure 4, please show with and without comparison so that readers can see how well the proposed method helps our visual evaluation. The same applies to Figure 5 and 6.
· Figure 7, I have no idea how to read this plot or what it is.
· Overall, the writing is a bit rough. I do not dare to list all the funny points, and the abovementioned is not an elaborate list. I expect the authors read the manuscript again to improve general clarity.
· All of these criticism said, again I want to encourage the authors that I like the basic idea of the manuscript. I expect the authors provide a good prove that a simple method goes well with our intuition and it works just as good as expensive methods.
Round 2
Reviewer 1 Report
thanks for your answers.
Reviewer 2 Report
Congratulation for your work. It is a pleasure to know authors who publish their algorithms. Congrats again.
Reviewer 3 Report
Sensors 1740238.R1
The authors should be commended for their revision of their manuscript, and broadly I think the manuscript has been improved. However, I still have a number of major concerns regarding the work.
1. In my first review, I asked the reviewers to discuss the precision of their system. In their response to this comment, the authors write: “There is little to discuss unless we change our approach or add more steps to the signal processing and cleaning. However, this would defeat the rationale for this work, as it will tend to produce a dataset-tailored approach like the classification.”
a. While I take their point that the results of their system are the results as there is no iteration within their pipeline. I do not agree with the statement that this means that there is nothing to discuss. At the very least, the authors need to acknowledge that a precision value of 50% is simply not good, and the variance in precision between individuals for a dataset driven approach is meaningful. I would further suggest that the authors discuss options for improving precision, and what changes/additions to their pipeline that could be made .
2. Furthermore, the editing in the manuscript has tightened up the study rationale, but it is still fairly limited. There a number of responses from the authors to the comments from the previous review that could be included in the manuscript to strengthen and/or provide additional relevant detail.
3. Finally, despite the editing of the authors, a number of the specific details of the proposed system remain somewhat unclear i.e., what is the normalization step from Figure 2?. I would urge the authors to be more explicit in their descriptions and/or even consider including relevant portions of their code.
A number of spelling mistakes, and grammatical and syntactical errors that are most likely the result of the editing process remain, but they currently do not overly hamper the reading of the manuscript.
My specific comments are as follows:
1. In their response, the authors write: “Given the cost of expert labour and the large testing volume, an hour of expert review of a single recording is very unproductive. In addition, the promise of new electrode technology for ultra-long-term EEG recordings and current requirements for real-time review has created a desperate need for automated EEG review. This method can make significant improvements in productivity and lower the costs of testing.”
a. This information (or some form of it) needs to be directly included in the manuscript and cited. It provides salient information to the study rationale that is otherwise not raised.
b. Similarly the authors raise the point of cost and time of visual inspection, I would suggest consider including specific information. How costly, give an estimate of both time and monetary value.
2. I’d be inclined to not use the phrase “baseline drift” to describe the presentation of the seizure. Baseline drift is already a term used in EEG work, and actually used by the authors in describing their filtering steps.
3. As I stated in my initial review, the authors need to provide the specific details of their chosen filters. Using descriptors like “excellent transition band features” is not replicable.
4. In their edit the authors have now described the first filter applied in the pipeline as a low-pass FIR filter, rather than the formerly described band-pass FIR filter. Has this made any difference to their data and results?
5. In the manuscript, the authors still do not address the paediatric nature of their sample, and how neural development and changes in the presentation of epilepsy as children age may affect their system.
a. There are a number of sections of their reply that could be transitioned into the manuscript to directly address these points.
6. In my original review, I asked about the 7 patients who had partial-onset seizures, and patient-specific true/false positive/negative rates, and if type of seizure had an impact. The authors state that this is beyond the scope of the manuscript, however, I disagree. If the purpose of the manuscript is to present the viability of the system, then these specific information contribute towards that purpose.
7. In their response, the authors describe that they used a filter of length 50 (for which the assumable unit is data points/samples), but in the manuscript they describe an order of 50. Typically Order is equal to the filter Length – 1, the authors should clarify.
8. The more detailed information contained in the response letter concerning the 2-Zeros 2-Poles filter needs to be included in the manuscript. More specifically, the information regarding the central frequency of the seizure band.
9. The authors have added a paragraph describing using “several parameters” to merge together contiguous seizure intervals. These parameters and their utilisation need to be explained in more detailed, as does the rationale for removing the intervening signal.
10. Following this paragraph, the authors write: “The seizure algorithm mentioned here uses only a simple logical analysis based on the absolute value of the denoised and filtered data.”
a. What exactly is the logical analysis? The following lines appear to describe this from a more qualitative space, but the pipeline appears to use a quantifiable method.
b. Perhaps the authors can better describe the phrase “discontinuity in the DC level”, from their description of a void space it appears to refer to a movement away from the 0 line, but this is unclear.
11. Largely the figures and tables need either more detailed and descriptive captions, or to be more detailed. At the moment a number of them would not be able to be read without the accompanying text.
12. Despite the authors comment regarding this being challenging due to size, Figure 6 really needs to better identify the ‘set threshold’ so as to be able to see how channels 1, 5, 9, and 14 are beyond this. At the moment it is visually unclear as to why these channels and not others (e.g., Chan 16, and 21) meet the threshold.
a. Perhaps consider an image inset that zooms into say Chan 1 and Chan 2 and demonstrate how one channel crosses the threshold and another does not.
13. Table 2 is unclear; the authors are just presenting a series of file names and what appears to be the temporal location of the seizure inside the recording. How exactly does this indicate any important information?
14. Figure 7a can have its X axis constrained to 25 rather than 30. It will allow the data points to spread out more and be more visible.
15. The captions/titles for Figure 7a, 7b, and 7c are not informative and do not meaningfully describe the data contained in the figure.
16. I would like to see Table 5 include hemispheric information where possible (i.e., the 7 patients with partial seizures)
a. Further, I previously asked about these regions, and the authors noted that there are no detailed annotations (assumedly regarding the seizure location). These information need to be included in the description of the table.
17. Table 7 needs to include the results of the present work to be a meaningful comparison. Otherwise, it’s more of an introductory item and should be introduced far earlier in the manuscript.
18. Figure 8 is unclear, and its axes/data are not explained in-text. As I mentioned in my previous review, what exactly does the label “number of trials” refer to? Similarly, what are the units for rate of performance? This information needs to be included in the figure and/or it’s caption
19. The author’s response to my previous comments 26 and 30 each need to be included in some degree in the discussion section of the manuscript.
Reviewer 4 Report
The authors responses are mostly ok. I did not appreciate the authors attached directly edited Word file with all the histories on, which was very distracting. Next time, please do not do this.
I must say that the authors’ response below was terrible and definitely made their impression bad.
The 2-poles 2-zeros filter is a special filter sometimes called a “resonator” designed to elicit resonance at specific frequencies. In the past, it was used to sync a receiver to a specific carrier (central frequency), and its tailored narrow pass-band ensured selectivity of information e.g. specific radio stations. In this analogy, we are treating the central frequency of the seizure as the carrier. This approach has been proved successful in other physiological “spiky” features detection e.g. QRS detection, where a frequency dependant derivative filter (zeros) is firstly used to enhance the change points of the targeted features and then dominated by the application of related integrative filters (poles).
Now I doubt if the authors really know what they are doing. In my previous comment, I specifically requested that I wanted to see the frequency profile of the 2-Zeros 2-Poles bandpass FIR filter. It was because I wanted to know the pass band of the filter and the depth of the suppression etc.. If the authors really know what they are doing, they should be able to show the frequency-domain response of the filter. In EEG analysis, pass/stop-band frequencies, as well as the suppression depth, slope, and ripple amplitudes are the most important. Without reporting these parameters, the authors are not explaining anything important about the filter. It is true that showing the corresponding 2-D Pole-Zero plot can substitute for this description, but the IIR notch filter is not explained in that way. These descriptions should be consistent in the paper, and again I request the authors either show or report the parameters of the bandpass FIR filter. This band-pass filter is a part of key solutions of the proposed idea hence it must be clearly described.
Round 3
Reviewer 3 Report
The authors have made a strong attempt at revising their manuscript, particularly focusing on detailing their proposed method; it is considerably easier to understand and most likely replicate now. For that the authors should be commended.
My remaining major comment is that a number of the newly added sections require additional detail regarding a number of newly named parameters/processing steps. I would suggest the authors revise these sections to ensure that all parameters can be understood in terms of how they are calculated, what they represent, and how any adjustments to these parameters could change the algorithm output.
I would suggest a final grammatical and syntactical edit just to catch any mistakes that are most likely the result of the revisions.
My specific comments are as follows:
1. In response to my previous comments, the authors have added some notes regarding the cost of expert labor, which they approximated to $150 USD an hour (the estimated time for visual examination). While it is a cost, I do not know if I would say that it is particularly expensive (especially in a space like healthcare) and use that as a key component of my rationale. I think the authors would be better to rely more on their point about ultra-long recordings, data size, and more real-time review.
a. If the authors feel inclined to retain this statement, I would suggest they unify the costs with the prevalence statistics i.e., if reporting on Australian incidence report on Australian costs.
2. What was the rationale for the recurring notch filter to reduced DC offset? Why not just subtract the mean voltage value (i.e., the DC offset) as is common with EEG analyses?
3. In their response, the authors state that when examining the recordings of the patients with low precision, they noted high levels of artefacts in those recordings, and that they recognise that they must run physical artefact removal prior to their algorithm.
a. Has this information been included in the manuscript? Because realistically you are now proposing a processing pipeline that happens to include the algorithm of the authors.
b. To that end, what artefact detection/removal method/s are suitable/compatible with the proposed algorithm?
4. The authors now include an introduction of the conversion of multi-channel data to single channel, and that this ‘masks’ the original EEG signal. It’s an awkward inference, and implies that the original data is being hidden.
a. I would suggest they rewrite this in the positive i.e., the algorithm allows the features of interest to be extracted from the original signal.
5. The description of the pre-processing steps (i.e., Section 2.3) still uses the phrase “excellent transition band features”. As per my previous review, please describe these.
6. In response to my 4th comment on the changed nature of the first filter (from a band-pass FIR filter to a low-pass FIR filter) the authors provide a description of “The Feature Filter” and how this intersects with the pre-processing FIR filter.
a. While important to the manuscript, it’s not really relevant to my question. My question was more centered on why the type or description of the filter changed.
b. I think the authors should clarify in their response the exact nature and type of all filters and implemented processing steps.
7. Section 2.5 (and the authors response to my previous comment 9) describes a series of variables that appear to have “ideal” values or bounded values (i.e., “The lowest limit is usually between 0.05 and 0.1 and its upper limit is between 0.8 and 1”). These values appear to be fundamental to the determination of a seizure, but neither their calculation nor nature is described.
a. What would changing these values do to the detection procedure? Do they need to be calculated per cohort, per individual, etc.
8. In Sections 2.7 and 3, the authors describe duration bounds (of less than 0.4 seconds, and greater than1.8 seconds), which excludes a discontinuity being counted a seizure. Where did these parameters come from, and how were they derived?
9. The authors have responded that they have edited the Figure captions/titles to be more informative. But I am not sure what exactly has been edited, as most of the captions seem unedited (perhaps this is an issue with the compiling/submission software?)
a. That said, there are a number of descriptions in their response that contain pertinent information (response to comment 18) that should be contained as a note for these figures, if they have not already been.
10. There appear to be a few missing words in Section 4. For example, “The lack of various datasets annotated with seizures for adults make the extensive validation of our algorithm. Overall,”
a. Makes the validation of the algorithm what?
11. In line with a previous comment, in Section 4 the authors write: “EEG signals from patients 6 and 13 contain significant artefacts, and an improved pre-processing method is required to distinguish ictal events.”
a. What does this look like? Provide details
12. Again, in line with a previous comment, in their conclusion the authors write: “…requiring tailored adjustment of the algorithm parameters (pks, locs, width in Matlab code).”
a. This presents a fundamental problem for the current work. A lot of the rationale is built on the difficulties of patient-specific nature of existing automated classifiers. And now here the authors are also suggesting patient-specific adjustments.
b. How then is the proposed system better if it still faces the same problem? The authors need to discuss the implications of these final statements, beyond saying that manual processing is an alternative and that more work on the algorithm is required.
Reviewer 4 Report
See the attached Word file for my comments, figures for the results from replication, and Matlab code used to run the replication.
